# Rapid Responses of Greenhouse Gas Emissions and Microbial Communities to Carbon and Nitrogen Addition in Sediments

**DOI:** 10.3390/microorganisms12101940

**Published:** 2024-09-25

**Authors:** Jin-Feng Liang, Bo Yao, Xiao-Ya Zhang, Qi-Wu Hu

**Affiliations:** 1School of Geography and Environment, Key Laboratory of Poyang Lake Wetland and Watershed Research, Ministry of Education, Jiangxi Normal University, Nanchang 330022, China; 2College of Ecology and Environment, Hainan Tropical Ocean University, Sanya 572000, China

**Keywords:** glucose addition, nitrate nitrogen, greenhouse gas emission, functional gene

## Abstract

Massive labile carbon and nitrogen inputs into lakes change greenhouse gas emissions. However, the rapid driving mechanism from eutrophic and swampy lakes is not fully understood and is usually contradictory. Thus, we launched a short-term and anaerobic incubation experiment to explore the response of greenhouse gas emissions and microbial communities to glucose and nitrate nitrogen (NO_3_^−^-N) inputs. Glucose addition significantly increased CH_4_ and CO_2_ emissions and decreased N_2_O emissions, but there were no significant differences. NO_3_^−^-N addition significantly promoted N_2_O emissions but reduced CH_4_ accumulative amounts, similar to the results of the Tax4Fun prediction. Bacterial relative abundance changed after glucose addition and coupled with the abundance of denitrification genes (*nirS* and *nirK*) decreased while maintaining a negative impact on N_2_O emissions, considerably increasing methanogenic bacteria (*mcrA1*) while maintaining a positive impact on CH_4_ emissions. Structural equation modeling showed that glucose and NO_3_^−^-N addition directly affected MBC content and greenhouse gas emissions. Further, MBC content was significantly negative with *nirS* and *nirK*, and positive with *mcrA1*. These results significantly deepen the current understanding of the relationships between labial carbon, nitrogen, and greenhouse emissions, further highlighting that labile carbon input is the primary factor driving greenhouse gas emissions from eutrophic shallow lakes.

## 1. Introduction

In recent decades, human activities have led to a large amount of reactive nitrogen entering aquatic environments (such as shallow lake ecosystems), and excess nitrogen input as a pollutant would cause some irreversible ecological problems, such as lake eutrophication and swampiness, etc. [1,2]. The nitrogen trapped by shallow lakes could impact element cycling by changing the soil’s physicochemical properties, microbial communities, and labile carbon and nitrogen content [3,4,5], thereby affecting greenhouse gas emissions [6,7,8]. Moreover, nitrogen-rich and anaerobic environments are more conducive to greenhouse gas emissions [9].

Labile carbon and nitrogen mainly include ammonium, nitrate, dissolved organic carbon, soil microbial biomass carbon, and nitrogen. Nitrate (NO_3_^−^) is the main form of nitrogen in freshwater environments, for which removal in the lake sediments is dominated by denitrification [10], a major biological nitrogen loss process from natural ecosystems to the atmosphere, and contributes more than 70% of nitrogen loss as the form of nitrous oxide (N_2_O) or N_2_ [11,12]. Dissolved organic carbon (DOC) is usually abundant and mobile, and usually plays an essential role in biogeochemical cycles [11,13,14]. Generally, labile carbon and nitrogen (e.g., NO_3_^−^-N and DOC) are important environmental factors that influence the critical nitrogen cycle process (such as denitrification) and carbon cycle process, which further affects the dynamics of greenhouse gas emission in an anaerobic wetland system [15,16,17]. For instance, nitrogen addition led to an increase in N_2_O emission, while carbon addition significantly reduced N_2_O emission [18]. Research showed that DOC and inorganic nitrogen (NH_4_^+^-N and NO_3_^−^-N) as the crucial factors within lake carbon cycling, which are the essential substrates and resources for microorganisms associated with carbon dioxide (CO_2_) and methane (CH_4_) production [13,19]. Therefore, labile carbon and nitrogen content and form are essential in regulating the carbon and nitrogen cycle in anaerobic environments. 

As a crucial part of aquatic ecosystems, microorganisms play an important role in biogeochemical cycles by fixing and remineralizing nutrients and play a critical role in nitrogen and carbon cycles in the sediments [20,21]. External carbon and nitrogen inputs impact bacterial community compositions [22]. The changes in microbial communities that respond to variations in carbon and nitrogen further impact greenhouse gas emissions [23,24]. For example, Flavobacteria was enriched by nitrates in the environment [4]. Pseudomonas were significantly negative with N_2_O emissions, and Actinobacteria was positively associated with CO_2_ emissions [7,24]. Furthermore, soil microbial biomass is a sensitive measure of microbial activity, and microbial biomass influences greenhouse gas emissions by changing the substrate content for organic matter mineralization and methanogenesis [25]. Microbial functional genes that encode nitrogen and carbon transformation enzymes are widely used as gene markers to represent denitrifies and anaerobic ammonium oxidation, thus playing a role in regulating N_2_O and CH_4_ production processes [26,27]. The net balance between the production and reduction of N_2_O emissions is mainly regulated by *nirK*, *nirS*, and *nosZ* genes, and CH_4_ emissions are mainly regulated by *mcrA* and *pmoA* genes [17,27,28]. Overall, these functional genes can provide an intrinsic view for understanding greenhouse emissions.

Thus, revealing the interactions between labile carbon and nitrogen and greenhouse gases will contribute to understanding the mechanism of greenhouse gas emissions from wetland ecosystems to the atmosphere, and elucidate the contribution of external carbon input on dynamic changes in nitrogen content in slightly eutrophic wetland ecosystems. Currently, major studies related to labile carbon and nitrogen influence on greenhouse gases mainly focus on field investigation, including inland wetlands, estuarine wetlands, and rivers [2,29,30], or relatively long-term incubation experiments with single carbon or nitrogen addition, and/or mainly focuses on single gas emissions [23,31,32]. Furthermore, results are often inconsistent due to wetland types, soil physicochemical properties, etc. The rapid driving mechanisms of greenhouse gas emissions (CH_4_, CO_2_, and N_2_O) and microbial communities under anaerobic sediments with labile carbon and nitrogen input are not well understood. Thus, we launched a short-term anaerobic incubation experiment to elucidate the effects of labile carbon and nitrogen variation in (1) greenhouse gas emissions, (2) dynamic of microbial community composition and related functional genes, and (3) whether the sediment nitrogen content could be reduced after external labile carbon input? Understanding the underlying mechanisms is essential for formulating appropriate measures to alleviate the greenhouse effect and nitrogen pollution in nitrogen enrichment lakes.

## 2. Materials and Methods

### 2.1. Sediments Sampling

We collected surface sediments (0–10 cm) from the site of Zaolinzhuang, which had a lower nitrogen content in the Baiyangdian Lake (38°43′–39°02′ N, 115°45′–116°07′ E) in Hebei Province, China (Figure 1).

Sediments were collected using a Van Veen grab sampler (Eijkelkamp, Giesbeek, The Netherlands) at a depth of approximately 10 cm. After collection, sediments were brought to the laboratory as soon as possible, and removed the litter, gravel, and coarse debris manually. The sediment samples were divided into three parts. One part was stored at 4 °C for the determination of dissolved organic carbon (DOC) and inorganic nitrogen; one part was air-dried and sieved (≤2 mm) for determination of total carbon (TC), total nitrogen (TN), pH, EC, and particle size (Clay, Silt, and Gravel), and the remaining part was used for the incubation experiments. The methods of ammonia nitrogen (NH_4_^+^-N), nitrate nitrogen (NO_3_^−^-N), and DOC contents were similar to our previous study [33]. TC and TN contents were measured by an Elemental analyzer (Vario EL III, Elementar, Frankfurt, Germany). pH and EC were measured by the water leaching potentiometric method (sediment/water ratio 1:5, *w*/*v*) using a pH meter (PB-10, Sartorius, Gottingen, Germany) and a salinity meter (FE38, Mettler Toledo, Oakland, CA, USA), respectively. Soil particle size distribution was analyzed on a laser particle size analyzer (Microtrac S3500, Microtrac Inc., Largo, FL, USA). The physicochemical properties of the dry mass of sediments are shown in Table 1.

### 2.2. Experimental Design and Incubation

The experiment was complete factorial designed with three NO_3_^−^-N addition treatments (no addition, 0 mg kg^−1^—control; 10 mg kg^−1^—low; 20 mg kg^−1^—high; dry mass), and two carbon addition treatments (glucose addition and no addition as a control). Making a total of 6 treatments and with 3 repetitions per treatment. For each treatment, the dry sediment (50 g) was put into the 250 mL bottles, and NO_3_^−^-N additions were added as a dissolved solution, then we added 0 mg, 3.606 mg, and 7.212 mg KNO_3_ to 30 mL deionized water and poured into the bottles, named as the control, low, and high nitrogen level treatments, respectively (Figure 2). Before being sealed with a rubber plug, the headspace of the bottles was purged with helium gas to maintain anaerobic conditions. Then, a 3-day pre-incubation was conducted at 20 ± 0.2 °C in the dark to activate microbial activity. After pre-incubation, to the half bottles was added 20 mL dissolved glucose solution (50 mg glucose, additive amount based on 2% SOC contents); 20 mL deionized water was directly added to the other half bottles as the control. Further, all bottles were flushed with helium gas until an anaerobic environment was formed. Additionally, there were three empty bottles as a blank control for greenhouse gas analysis. All bottles were incubated at 20 ± 0.2 °C in the dark under anaerobic conditions for 7 days.

### 2.3. Measurements

#### 2.3.1. Greenhouse Gas Collection and Measurement 

The headspace gases in the bottles were sampled with a 20 mL airtight syringe at 1, 2, 3, 5, 6, and 7 days, respectively. The bottles were purged with helium and sealed after each collection. Each gas sample’s CO_2_, CH_4_, and N_2_O concentrations were determined using a gas chromatograph (Agilent 7890B, Santa Clara, CA, USA) equipped with an electron capture detector (ECD) immediately. Emissions rates of CO_2_, CH_4_, and N_2_O were calculated using the following equation [34]:$$E=P\times V\times \frac{\Delta c}{\Delta t}\times \frac{1}{RT}\times M\times \frac{1}{m}$$
where *E* stands for greenhouse gas emission rate (ng g^−1^ h^−1^), *P* and *V* refer to standard atmospheric pressure (Pa) and headspace volume of the bottle (cm^3^), respectively, *c* indicates the concentration of CO_2_, CH_4_, and N_2_O (ppb), *t* is time among sampling interval (h), *R* means the universal gas constant, *T* is on behalf of absolute air temperature (K), *M* refers to molecular mass of CO_2_, CH_4_, and N_2_O (g mol^−1^), and *m* is incubated sediments mass according to dry weight (g).

Cumulative emission of CO_2_, CH_4_, and N_2_O were calculated as:$${GHG}_{cumlative}=\sum_{i=1}^{n}\frac{F_{i+1}+F_{i}}{2}(t_{i+1}-t_{i})$$
where *F*_i+1_ and *F*_i_ are gas emission flux at time *t*_i+1_ and *t*_i_, respectively [35].

#### 2.3.2. DNA Extraction, PCR Amplification, and Metagenomic Sequencing

We collected the solution overlying sediments and sediments in each bottle at the end of the experiment. Solution and partial sediment samples were stored at 4 °C to determine inorganic nitrogen and DOC contents. Another partial sediment was stored at −20 °C to determine the microbe. Then, 0.5 g sediment was used to extract total microbial DNA in the sediment using the E.Z.N.A.^®^ soil DNA Kit (Omega Bio-tek, Norcross, GA, USA) according to the manufacturer’s instructions. The concentration and purity of extracted DNA were determined with TBS-380 and NanoDrop2000, respectively. The DNA extract was checked on 1% agarose gel. Universal primers 338F (5′-ACTCCTACGGGAGGCA GCAG-3′) and 806R (5′-GGACTACHVGGGTWTCTAAT-3′) specific to the V3 to V4 region of the bacterial 16S rRNA gene were used for PCR amplification. PCR was performed in triplicate 20 mL reactions. The PCR protocol was 94 °C for 5 min, 30 cycles of 94 °C for 30 s (denaturation), 54 °C for 30 s (annealing), 72 °C for 45 s (extension), and 72 °C for 10 min. All samples were performed according to formal experimental conditions, the PCR amplification products were detected by 1.5% Invitrogen agarose, and the PCR products were purified using AMPure Beads (Beckman Coulter, Inc., Brea, CA, USA). The purified products were then stored at −20 °C and transported to Majorbio Biopharm Technology Co., Ltd. (Shanghai, China) for Illumina MiSeq sequencing (Illumina Inc., San Diego, CA, USA) [36]. 

#### 2.3.3. Sequence Quality Control and Genome Assembly

The raw reads from metagenome sequencing were used to generate clean reads by removing adaptor sequences, trimming, and removing low-quality reads (reads with N bases, a minimum length threshold of 50 bp, and a minimum quality threshold of 20) using the fastp on the free online platform of Majorbio Cloud Platform (cloud.majorbio.com). These high-quality reads were then assembled to contigs using MEGAHIT (parameters: kmer_min=47, kmer_max=97, step=10) (https://github.com/voutcn/megahit, version 1.1.2), which uses succinct de Bruijn graphs. Contigs with a length of over 300 bp were selected as the final assembling result.

#### 2.3.4. Processing of Sequencing Data

The raw 16S rRNA gene sequencing reads were demultiplexed, quality-filtered by fastp version 0.20.0, and merged by FLASH version 1.2.7. Operational taxonomic units (OTUs) with 97% similarity cutoff were clustered using UPARSE version 7.1. The taxonomy of each OTU representative sequence was analyzed by RDP Classifier version 2.2 against the 16S rRNA database (Silva v138) using a confidence threshold of 0.7. Representative sequences of non-redundant gene catalog were annotated based on the NCBI NR database using blastp as implemented in DIAMOND v0.9.19 with e-value cutoff of 1×10^−5^ using Diamond (http://www.diamondsearch.org/index.php, version 0.8.35) for taxonomic annotations. The KEGG annotation was conducted using Diamond against the Kyoto Encyclopedia of Genes and Genomes database (http://www.genome.jp/kegg/, version 94.2, accessed on 1 December 2023) with an e-value cutoff of 1 × 10^−5^. 

### 2.4. Statistical Analysis

The differences in sediment carbon and nitrogen contents, emission rate, and cumulative amounts of CO_2_, CH_4_, and N_2_O among all samples were evaluated by two-way analysis of variance (ANOVA) with a significance level of 5%. The statistical tests were performed using the SPSS 18.0 software package (SPSS Inc., Chicago, IL, USA). The function prediction of Tax4Fun was based on the conversion of the 16S rRNA database to prokaryotic taxonomy in the KEGG database. All sequencing data were analyzed using the Majorbio I-Sanger Cloud Online Platform. Linear regression analysis examined the relationship between MBC and *nirS*, *nirK*, *nosZ*, *mcrA*1, *mcrA*2, and *pmoA*. Structural equation models (SEM) were built to explain the mechanism by which various factors influence accumulative greenhouse gas (CO_2_, CH_4_, and N_2_O) emissions.

## 3. Results

### 3.1. Nitrogen and Carbon Contents in the Sediments

Nitrogen addition significantly affected MBC content in the sediments (Table 2, *p* < 0.05), and MBC content was significantly higher in the high nitrogen level than in others (Figure 3a,c). Compared with control (no glucose addition), glucose addition significantly decreased the NO_3_^−^-N content by 12.2%, and significantly increased MBC content by 90% after glucose addition (Figure 3a,c). Nitrogen level and glucose addition had significant interactive effects on MBC content (Table 2, *p* < 0.05). The highest MBC content was in the high nitrogen level with glucose addition (Figure 3c). Further, the glucose addition had no significant effects on NO_3_^−^-N content in the solution, and the lowest value was at the high nitrogen level (Appendix A, *p* < 0.05). DOC content in the solution significantly increased after glucose addition (Appendix A, *p* < 0.05).

### 3.2. Greenhouse Gas Emission after Glucose Addition 

Compared with the control, glucose addition significantly increased the emission rates of CH_4_ and CO_2_ (Table 3, *p* < 0.05; Figure 4a,b), while the N_2_O emission rate generally declined after glucose addition, even though there were no significant differences (Table 3, *p* > 0.05). NO_3_^−^-N addition significantly increased the N_2_O emission rates, and the maximum N_2_O emission rate was observed at the high nitrogen level (Table 3, *p* < 0.05; Figure 4c). In particular, NO_3_^−^-N addition significantly decreased the CH_4_ emission rates at day 2 and day 3 (Appendix A, Figure 4a). During the incubating period, the trend of N_2_O emission rates was maximum at day 1, reached 4717.7 μg g^−1^ day^−1^, and then rapidly decreased (Figure 4c). The CH_4_ and CO_2_ emissions trends were similar and maximum at day 6, which were 19.3 μg g^−1^ day^−1^ and 3761.8 μg g^−1^ day^−1^, respectively (Figure 4a,b). 

Similarly, the cumulative CH_4_ emission was highest in the control compared with other NO_3_^−^-N addition treatments (Figure 5a); cumulative N_2_O emissions were significantly increased with the increasing NO_3_^−^-N addition, and highest in the high nitrogen level (Figure 5c). Glucose addition significantly increased the cumulative CH_4_ and CO_2_ emissions, and slightly reduced N_2_O emission but had no significant impact (Figure 5d–f). 

### 3.3. Bacterial Community Structure and Related Associated Function Genes in the Sediments 

Relative abundances of the bacterial community (top 8) at the phylum level are shown in Figure 6. Firmicutes (0.25–0.67), Proteobacteria (0.10–0.24), Chloroflexi (0.05–0.18), and Actinobacteria (0.04–0.13) were the dominant bacterial phyla in the sediments, which the percentages accounted for more than 75% of the bacterial sequences in all samples. Glucose addition significantly increased the relative abundances of Firmicutes, while the abundances of Proteobacteria, Chloroflexi, Actinobacteria, and Bacteroidetes significantly decreased at the end of incubation (Appendix A, *p* < 0.05), while they were not significantly affected by the nitrogen level (Appendix A, *p* > 0.05). Functional predictions of Tax4Fun indicated that the abundances of denitrification, nitrite denitrification, and nitrate denitrification were decreased after glucose addition, fermentation, and chemoheterotrophy functions were increased after glucose addition, which was consistent with the trends of greenhouse gas emissions (Appendix A). Accompanied by RPKM, the abundance of denitrification functional genes *nirS* and *nirK* significantly decreased after glucose addition, and *nosZ* was significantly reduced (Table 4, *p* < 0.05; Figure 7a–c). Glucose addition also obviously increased the abundance of methanogen genes *mcrA*1 and decreased *pmoA* at low and high nitrogen levels, thus promoting CH_4_ emissions (Table 4, *p* < 0.05; Figure 7d,f). 

### 3.4. Relationships among Greenhouse Gas, Soil Nutrients, and Microbes 

The SEM showed that glucose addition had a significantly positive effect on CH_4_ and CO_2_ emissions with the path coefficients at 0.83 and 0.95, respectively (Figure 8a,b, *p* < 0.05), while not significantly decreased the N_2_O emission (Figure 8c, *p >* 0.05). Nitrate nitrogen addition significantly decreased the emissions of CH_4_ and CO_2_ with the path coefficients at −0.29 and −0.17, respectively (Figure 8a,b), but greatly promoted the N_2_O emission. Glucose addition and nitrogen addition explained 73% of the variance in MBC (path coefficient = 0.80 and 0.29, respectively), but MBC and sediment NO_3_^−^-N contents did not significantly affect CH_4_, CO_2_, and N_2_O emissions (Figure 8, *p >* 0.05). 

Based on the results of Figure 8, we found that the MBC content was significantly affected by the external labile carbon and nitrogen. Thus, we selected it as our study’s most important biological factor. Regression analysis showed that MBC can significantly negatively affect the relative abundance of *nirS* and *nirK* (Figure 9a, *p* < 0.05), and positively affect *mcrA* (Figure 9b, *p* < 0.05).

## 4. Discussion

### 4.1. Impact of Labile Carbon and Nitrogen Inputs on Greenhouse Gas Emissions

Nutrients, such as labial carbon and nitrogen, directly influence greenhouse gas emissions [37]. Usually, NO_3_^−^-N is the direct substrate for denitrification, and the carbon source is a necessary energy material for denitrifying bacteria [38]. The N_2_O emissions were significantly increased with the increased external nitrogen input, but the NO_3_^−^-N content in the sediments had no significant differences among nitrogen levels in this study, while the NO_3_^−^-N content in the solutions was lowest in the high nitrogen level, which may explain that the externally added NO_3_^−^-N mainly existed in solution and was used for denitrification. Thus, less NO_3_^−^-N exists in the high nitrogen level compared with other nitrogen treatments. Previous studies have shown that decreased NO_3_^−^-N content and increased DOC content may suppress the N_2_O emissions [18,39], and this phenomenon was partially consistent with our results that glucose addition only slightly decreased N_2_O emissions in this short-term study. Additionally, SEM showed glucose addition significantly decreased sediment NO_3_^−^-N content, but the NO_3_^−^-N content did not directly and significantly affect N_2_O emission. Glucose as a labile carbon source has a significant promoting influence on CO_2_ and CH_4_ productions in this study, also reflected by the results of SEM with the path coefficients at 0.73 and 0.84, respectively. The increased relative abundance of *mcrA*1 after glucose could also explain the increased CH_4_ emissions, as verified by the results that carbon content was positive with CO_2_ and CH_4_ emissions [28,37]. DOC regulates carbon availability and further affects soil microbial activities [25]. Therefore, higher DOC content could stimulate bacteria that are responsible for organic matter decomposition and methanogenesis, leading to promoting CO_2_ and CH_4_ emissions [40], because CO_2_ emissions mainly come from soil respiration and organic matter decomposition by microorganisms under anaerobic conditions [41]. 

MBC is the labile fraction of soil organic carbon and a sensitive measure of microbial activities [42]. MBC can act as a metabolism substrate for soil microbes and can sensitively affect the activity of functional microorganisms, resulting in promoting greenhouse gas emissions [25]. As such, greater MBC provides more biological residues as substrates for methanogens, which promotes CH_4_ emissions [43]. In our study, glucose addition significantly increased MBC and was accompanied by increased CO_2_ and CH_4_ emissions. The linear regression results indicate that MBC positively and significantly affected the relative abundances of the methanogenic gene *mcrA*. Thus, the increased MBC content promoted CH_4_ emissions, but a significant correlation between MBC and greenhouse gas emissions was not found. This may be attributed to increased MBC first affecting functional genes, then regulating greenhouse gases, which implies that MBC was not the direct indicator of the response of greenhouse gas emissions to carbon and nitrogen inputs in our study. 

Additionally, external nitrogen input typically accelerated MBC decomposition in anaerobic conditions [44], which was consistent with our results that MBC was higher in the high NO_3_^−^-N level than in others. Further, SEM also verified that external NO_3_^−^-N addition positively affected the MBC content. After glucose addition, the CH_4_ emission rate gradually decreased with the increased NO_3_^−^-N level. This may have been due to the indirect inhibition of methane oxidation by the increase in other oxidative reactions combined with nitrate ions [8]. The rates and amounts of N_2_O emission gradually increased with the external increased nitrogen input, which is consistent with other research that N_2_O emissions increased after nitrogen addition [7,45]. Elevated soil NO_3_^−^-N content typically results in higher N_2_O/N_2_ ratios during denitrification, probably due to the preferential use of NO_3_^−^-N as a terminal electron acceptor and N_2_O reductase activity inhibition by higher NO_3_^−^-N levels [46]. Further, external NO_3_^−^-N addition slightly decreased CO_2_ emissions, but there were no significant differences in this short-term study. A similar study showed that high nitrogen treatment inhibited CO_2_ emissions (33% reduction) in the same experimental plots, thereby increasing carbon sequestration [47]. Additionally, our previous study found a steady decrease in N_2_O emission rates after NO_3_^−^-N addition and kept stable after 7 days of incubation [45]. Therefore, we conducted this study for a short period of 7 days. 

### 4.2. The Changes in Soil Microbial Structure after External Nutrient Addition 

Microbial community structure is sensitive to resource availability, which can either be stimulated or inhibited depending on resource availability and supply, especially whether carbon and nitrogen are available [48]. Bacterial community compositions are significantly influenced by carbon content. Firmicutes were the dominant species and had the highest value in our study, followed by Proteobacteria, Chloroflexi, and Actinobacteria. This is consistent with the other research that the Firmicutes were predominant in all samples [49]. Usually, Firmicutes, Proteobacteria, and Bacteroidetes are mainly involved in the denitrification process in rivers and lakes [50]. A histogram indicated that glucose addition significantly decreased Actinobacteria, which indirectly indicated that external carbon input was beneficial in alleviating eutrophication in our study, because Actinobacteria were significantly positively correlated with nitrogen contents [36]. Chloroflexi were commonly found in the sediments and are considered one of the main bacteria that utilize easily decomposable carbon sources [51], but the abundance of Chloroflexi was decreased after glucose addition in our study. This can be explained by Chloroflexi, as an oligotrophic group, tending to be inhibited in response to organic substrate inputs [52]. Generally, nitrogen addition changes microbial community structure and diversity [3,53]. However, microbial community composition was not affected by the NO_3_^−^ addition, which may be attributed to the fact that when carbon and nitrogen are input together, microorganisms will preferentially utilize labial carbon sources. 

Usually, the denitrification process is affected by NO_3_^−^, organic carbon, and pH [54], consistent with a meta-analysis that N_2_O emissions were significantly increased after nitrogen input without a significant relationship with associated functional genes [55]. Previous research has documented that denitrification was widespread among Proteobacteria, Bacteroidetes, and Firmicutes [23]. These microbial communities were not significantly affected by the nitrogen levels, but they were significantly changed after glucose addition, and the accompanied denitrification (functional genes *nirS*, *nirK*, and *nosZ*) weakened. The results of functional predictions also showed that glucose addition affected the denitrification processes. A possible reason that can be given is that high carbon utilization may lead to anaerobic microsite formation and provide favorable conditions for heterotrophic denitrification [7]. In field experiments, nitrogen addition altered the abundance and diversity of nitrogen functional genes, including *nirK*, *nirS*, *nosZ*, etc. [7,26]. Regretfully, the functional genes related to denitrification were not affected after NO_3_^−^-N addition in our study, consistent with a meta-analysis that N_2_O emissions were significantly increased after nitrogen input without a significant relationship with associated functional genes [55]. Additionally, the function of fermentation and chemoheterotrophy were increased after glucose addition, which is responsible for the production of CH_4_ and CO_2_ [56]. The increased abundance of the methanogenic gene (*mcrA*) after glucose treatment may partially explain the increased CH_4_ emissions in our study and other studies [28]. This can be explained by the increased *mcrA* gene related to the consumption of DOC [28]. However, partial research has shown that DOC can limit the growth of methanogens and promote the utilization of methane by methanotrophs [57]. These findings indicate that the availability of carbon substrates might control CH_4_ emissions.

## 5. Conclusions

This study highlights the potential responses of nutrients, greenhouse gases, and microbial communities to external carbon input in eutrophic lakes. External carbon input was the dominant source of CH_4_ and CO_2_ emissions under ambient conditions, but the major contribution of denitrification genes (*nirS*, *nirK*, and *nosZ*) to N_2_O emission decreased and showed that N_2_O emissions decreased at the early stage of incubation. The NO_3_^−^-N pool was the dominant source of N_2_O, and increased NO_3_^−^-N addition was positive with N_2_O emissions but negative with CH_4_ emissions. MBC content significantly indirectly affects greenhouse gas emissions, which negatively with the abundance of *nirS* and *nirK*, and positively with *mcrA*1. Thus, labial carbon and NO_3_^−^-N played an important role in our study. Furthermore, appropriate carbon-to-nitrogen ratios play a key role in reducing greenhouse gas emissions and strengthening wetland carbon and nitrogen sequestration, and this needs to be considered in the future.

## Figures and Tables

**Figure 1 microorganisms-12-01940-f001:**
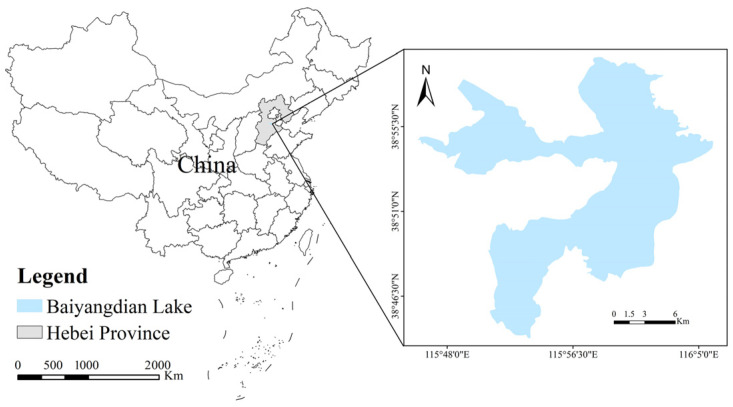
Geographic location of the sampling area.

**Figure 2 microorganisms-12-01940-f002:**
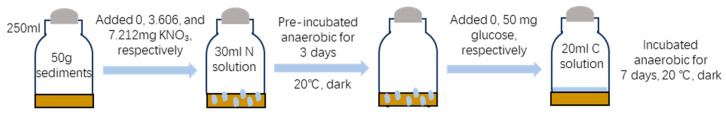
Diagram of experiment design.

**Figure 3 microorganisms-12-01940-f003:**
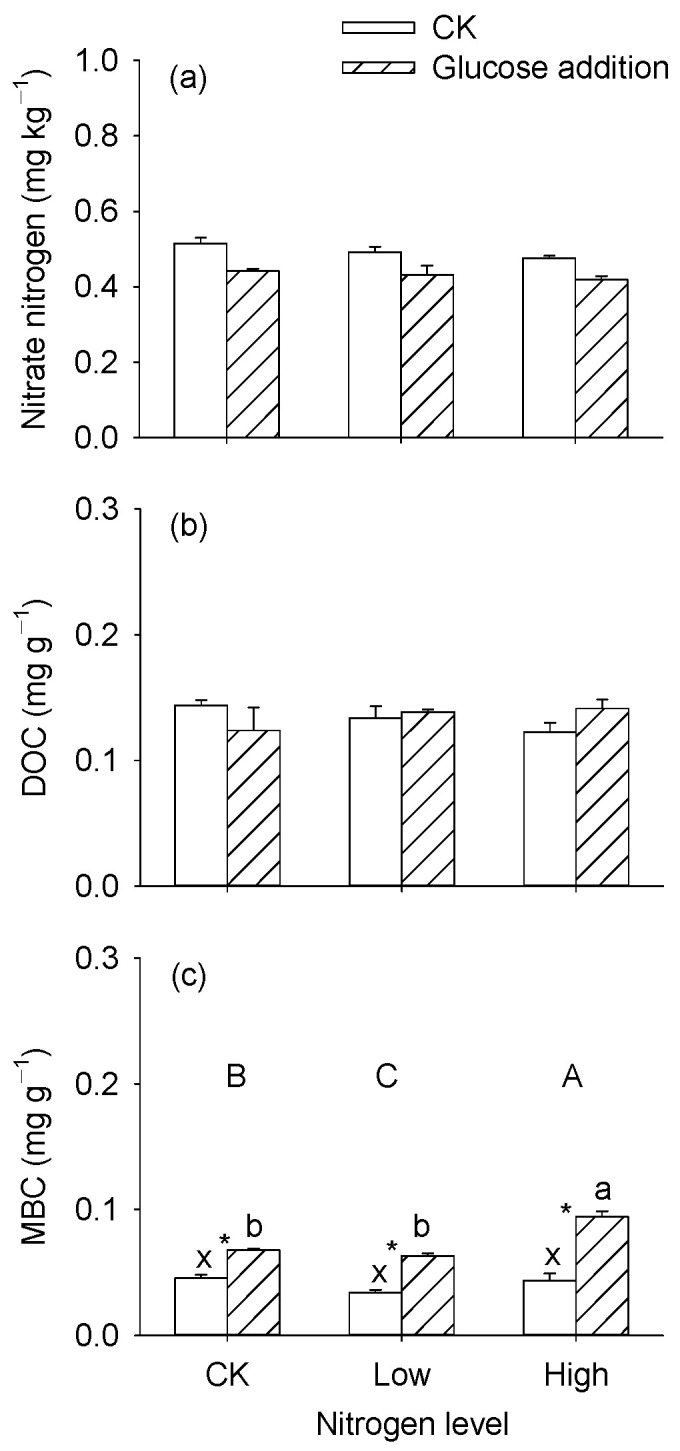
Effects of nitrogen and glucose addition on the nitrate nitrogen (**a**) and labile carbon contents (**b**,**c**). The contents refer to dry mass. Different capital letters indicate significant differences among nitrogen levels; within each glucose addition treatment, bars sharing the same lowercase letter are not significantly different at *p* = 0.05. Asterisks “*” show that means differ significantly between the control and glucose addition.

**Figure 4 microorganisms-12-01940-f004:**
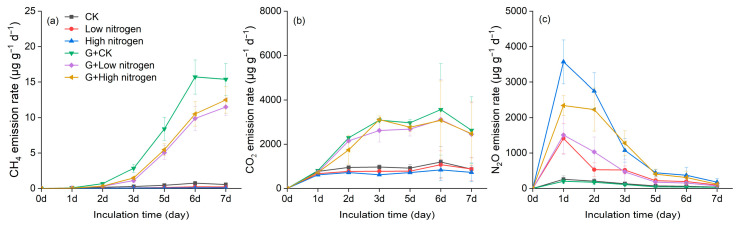
Effects of nitrogen and glucose addition on CH_4_ (**a**), CO_2_ (**b**), and N_2_O (**c**) emission rates.

**Figure 5 microorganisms-12-01940-f005:**
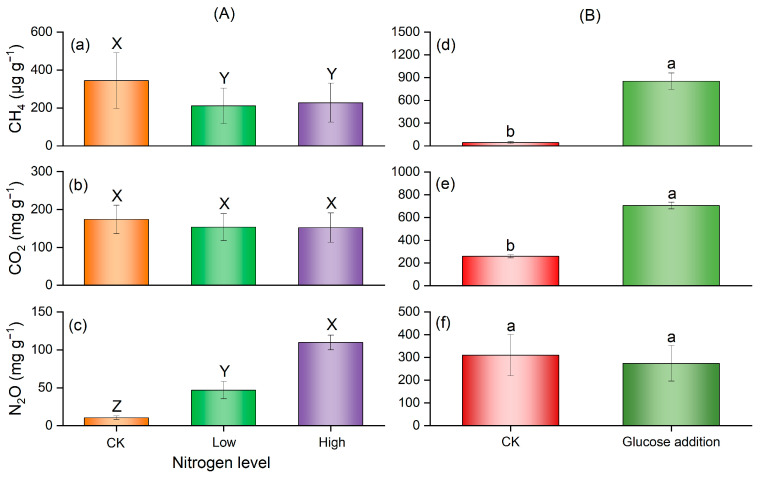
Effects of nitrogen and glucose addition on the cumulative emissions of CH_4_ (**a**,**d**), CO_2_ (**b**,**e**), and N_2_O (**c**,**f**). (**A**) The main effect of nitrogen addition; (**B**) The main effect of glucose addition. Different capital letters indicate significant differences among nitrogen levels; different lowercase letters indicate significant differences between the glucose addition treatments.

**Figure 6 microorganisms-12-01940-f006:**
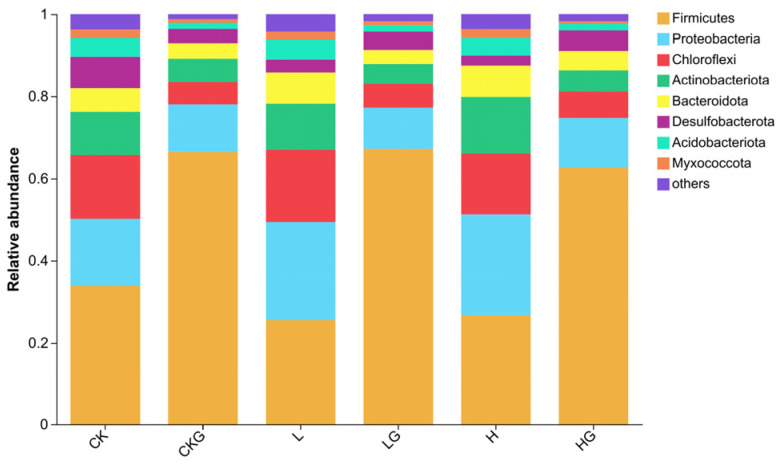
Effects of nitrogen and glucose addition on the relative abundance of bacterial community (Phylum level) in the sediments. Abbreviations of CK, CKG, L, LG, H, and HG represent the treatments of control, low, and high nitrogen levels without and with glucose addition, respectively.

**Figure 7 microorganisms-12-01940-f007:**
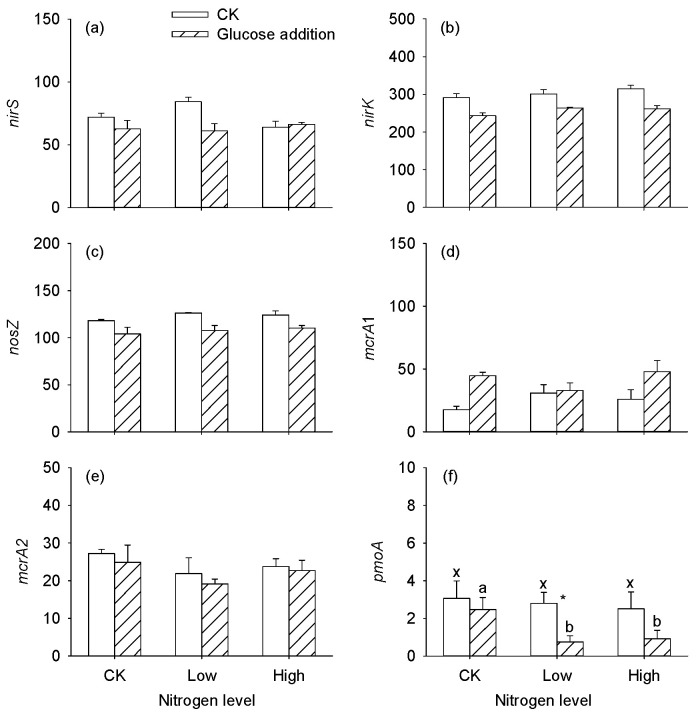
Effects of nitrogen and glucose addition on the relative abundance of denitrification functional genes (*nirS*, *nirK*, and *nosZ*; (**a**–**c**)), methanogen gene (*mcrA*; (**d**,**e**)), and methane-oxidizing gene (*pmoA*; (**f**)) based on RPKM. Within each glucose addition treatment, bars sharing the same letter are not significantly different at *p* = 0.05. Asterisks “*” show that means differ significantly between the control and glucose addition.

**Figure 8 microorganisms-12-01940-f008:**
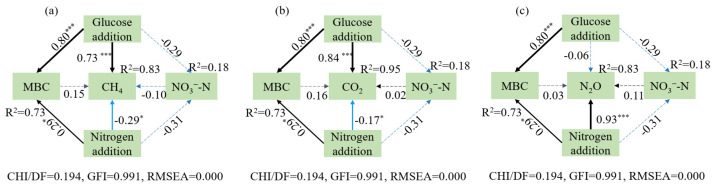
The structural equation models (SEM) analysis of the effect of nitrogen and glucose addition on the accumulative emissions of CH_4_ (**a**), CO_2_ (**b**), and N_2_O (**c**) via the pathways of NO_3_^−^-N and MBC. Black and blue solid arrows connecting the boxes represent significant positive and negative effects (*p* < 0.05), respectively. Asterisks indicate significance at *p* < 0.05, and *p* < 0.001 probability levels (*, and ***, respectively). Pathways without a significant effect are indicated by broken lines (*p* > 0.05). Values associated with the arrows represent standardized path coefficients. The R^2^ numbers denote the proportion of variance that could be explained by the corresponding variable in the structural equation model.

**Figure 9 microorganisms-12-01940-f009:**
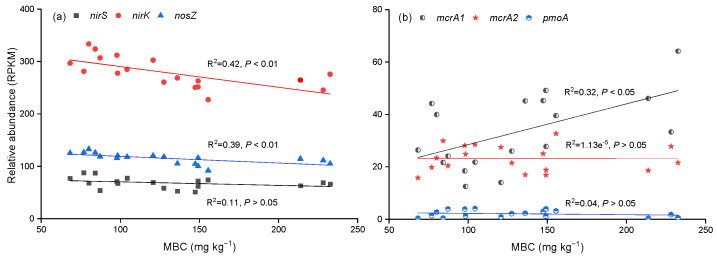
Linear regressions of MBC content with abundances of *nirS*, *nirK*, and *nosZ* (**a**) and *mcrA1*, *mcrA2*, and *pmoA* (**b**). n = 18.

**Table 1 microorganisms-12-01940-t001:** Physicochemical properties (mean ± SE, n = 4) of the sediments.

Variable	Sediments
Ammonia nitrogen (mg kg^−1^)	12.00 ± 6.01
Nitrate nitrogen (mg kg^−1^)	2.82 ± 1.26
Total nitrogen (mg g^−1^)	15.30 ± 0.31
Total carbon (mg g^−1^)	0.50 ± 0.02
Dissolved organic carbon (mg g^−1^)	0.11 ± 0.03
pH	7.62 ± 0.39
EC (μS cm^−1^)	560.33 ± 133.16
Clay (%)	3.74 ± 0.08
Silt (%)	29.74 ± 0.54
Gravel (%)	66.49 ± 0.62

**Table 2 microorganisms-12-01940-t002:** The ANOVA of nitrogen and glucose addition on the nitrogen and carbon contents.

Variables	Nitrogen (N)		Glucose (G)		N × G	
	*F*	*p*	*F*	*p*	*F*	*p*
Nitrate nitrogen (NO_3_^−^-N)	2.42	0.131	**29.01**	**<0.001**	0.35	0.713
Dissolved organic carbon (DOC)	0.09	0.917	0.02	0.883	1.70	0.228
Microbial biomass carbon (MBC)	**18.94**	**<0.001**	**157.97**	**<0.001**	**9.89**	**0.003**

Notes: The given are *F* and *p* of two-way ANOVAs. Values with *p* < 0.05 are in bold.

**Table 3 microorganisms-12-01940-t003:** The ANOVA of nitrogen and glucose addition on the greenhouse emissions.

Variables	Nitrogen (N)		Glucose (G)		N × G	
	*F*	*p*	*F*	*p*	*F*	*p*
(a) Emission rate						
CH_4_	1.21	0.302	**46.92**	**<0.001**	0.76	0.471
CO_2_	1.32	0.271	**148.78**	**<0.001**	0.141	0.869
N_2_O	**18.14**	**<0.001**	0.26	0.610	0.46	0.630
(b) Cumulative amount						
CH_4_	**4.01**	**0.046**	**85.00**	**<0.001**	2.32	0.140
CO_2_	2.15	0.159	**196.51**	**<0.001**	0.17	0.845
N_2_O	**35.60**	**<0.001**	0.48	0.502	0.99	0.397

Notes: The given are *F* and *p* of two-way ANOVAs. Values with *p* < 0.05 are in bold.

**Table 4 microorganisms-12-01940-t004:** The ANOVA of nitrogen and glucose addition on the relative abundance (RPKM) of functional genes.

Variables	Nitrogen (N)		Glucose (G)		N × G	
	*F*	*p*	*F*	*p*	*F*	*p*
*nirS*	1.50	0.262	**7.40**	**0.019**	3.86	0.051
*nirK*	2.75	0.104	**37.44**	**<0.001**	0.40	0.680
*nosZ*	1.34	0.298	**19.84**	**0.001**	0.19	0.827
*mcrA1*	0.51	0.613	**11.07**	**0.006**	2.18	0.156
*mcrA2*	1.74	0.218	0.72	0.414	0.04	0.959
*pmoA*	1.55	0.252	0.01	0.935	**3.96**	**0.048**

Notes: The given are *F* and *p* of two-way ANOVAs. Values with *p* < 0.05 are in bold.

## Data Availability

The data presented in this study are available on request from the corresponding author. The data are not publicly available due to privacy or ethical restrictions.

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
