# Peer review of "Rapid Responses of Greenhouse Gas Emissions and Microbial Communities to Carbon and Nitrogen Addition in Sediments"

_microorganisms, 2024, doi:10.3390/microorganisms12101940_

Round 1

Reviewer 1 Report

Comments and Suggestions for Authors

The authors conducted a short-term anaerobic incubation experiment to investigate the effects of glucose and nitrate addition on greenhouse gas emissions, microbial community composition, and related functional genes in sediments. They demonstrated that the addition of glucose significantly increased CH4 and CO2 emissions, decreased N2O emissions and NO3-N content, and influenced the composition of bacterial communities and functional genes related to denitrification and methanogenesis. The addition of nitrate increased N2O emissions but reduced CH4 accumulation. These results enhance the understanding of the relationships between readily available carbon, nitrogen, and greenhouse gas emissions. The manuscript is well-structured and organized. However, the work needs some enhancements before it can be published. I recommend revision of the manuscript based on the following comments:

*The paper lacks information on the dry mass content of the sediments used in the studies. Additionally, information regarding the content of organic and mineral dry mass would be helpful. Organic matter could have served as an additional source of carbon, which the authors accounted for in the control sample without glucose, but this should be discussed.

*The authors used an external source of organic carbon in the form of glucose. Therefore, presenting the results in relation to the mass of sediments may be misleading, as the sediments served as inoculum, and the amount applied may not have significantly influenced the processes. The impact was due to the glucose and nitrate load. This should be addressed in the discussion.

Below are additional comments related to the amount of sediments used, which will help understand the obtained results:

*The authors should add a diagram illustrating the course and main assumptions of the experiment, including incubation conditions, bottle volume, sediment-to-solution ratio, incubation time, etc. This would enhance the readability and interest in the article.

*Table 1. Are these values expressed per kg of dry mass of the sediments? This information should be included in the description.

*Section 2.2. Similar to Table 1. Are the nitrogen loadings expressed per kg of dry mass? If so, what was the moisture content of the sediments or what volume does 1 kg of dry mass of sediments occupy? This information would help better understand the obtained results.

*Line 117. "50 g air-dried sediment" what was the moisture content/dry mass of such a sample? This is crucial for understanding the obtained results.

*Figure 1. The authors express the concentrations of nitrate nitrogen, DOC, and MBC in mg/kg. It should be clarified in the figure legend what the mass refers to. The authors should consider adding graphs showing the effectiveness of denitrification or changes in TN and DOC concentrations in the solution. This would allow for a better understanding of what occurred during the incubation of the samples.

Reviewer 2 Report

Comments and Suggestions for Authors

The article entitled: "Rapid Responses of Greenhouse Gas Emissions and Microbial Communities to Carbon and Nitrogen Addition in Sediments" has potential for publication, after suggested revisions.

1) Authors should pay attention to the maximum number of words allowed in the abstract.

2) In the keywords, it is necessary to put a period (.) at the end of the last keyword.

3) The Introduction is well-founded scientifically; the authors used 29 references to write it.

4) In the methodology in item 2.1, I suggest that the authors include, in addition to the geographic coordinates, an image of the location of the Lake.

5) In Table 1, the EC variable has a very high standard deviation. Is this value correct? Please check.

6) Item 2.2 in line 119, check the SI unit.

7) The results are good but are somewhat scattered throughout the text. The example is discussed on page 5, line 203, figure 4d. However, figure 4d only appears on page 8. This hinders the reader's understanding. The authors need to review this arrangement.

8) I suggest increasing the size of Figure 2, as it is difficult to visualize the data contained in the graph.

9) The same situation in figure 6. Low visibility of the data.

10) I liked the conclusion, but I missed an indirect opinion from the authors.

11) Please check the references with the entire text. 57 references is a good number.

I look forward to the corrections and would like to see the work after the corrections.

Round 2

Reviewer 1 Report

Comments and Suggestions for Authors

The authors have significantly improved the manuscript. The article can be accepted for publication in its current form.

Reviewer 2 Report

Comments and Suggestions for Authors

After the requested corrections, the article can be accepted.